# Genome-Wide Identification and Expression Analysis of Wall-Associated Kinase (WAK) Gene Family in *Cannabis sativa* L.

**DOI:** 10.3390/plants11202703

**Published:** 2022-10-13

**Authors:** Hülya Sipahi, Terik Djabeng Whyte, Gang Ma, Gerald Berkowitz

**Affiliations:** 1Department of Agricultural Biotechnology, Faculty of Agriculture, University of Eskişehir Osmangazi, Eskişehir 26160, Türkiye; 2Agricultural Biotechnology Laboratory, Department of Plant Science and Landscape Architecture, University of Connecticut, Storrs, CT 06269, USA

**Keywords:** *Cannabis sativa* L., wall-associated kinase, gene expression

## Abstract

Wall-associated kinases (WAKs) are receptors that bind pectin or small pectic fragments in the cell wall and play roles in cell elongation and pathogen response. In the *Cannabis sativa* (Cs) genome, 53 CsWAK/CsWAKL (WAK-like) protein family members were identified and characterized; their amino acid lengths and molecular weights varied from 582 to 983, and from 65.6 to 108.8 kDa, respectively. They were classified into four main groups by a phylogenetic tree. Out of the 53 identified *CsWAK/CsWAKL* genes, 23 *CsWAK/CsWAKL* genes were unevenly distributed among six chromosomes. Two pairs of genes on chromosomes 4 and 7 have undergone duplication. The number of introns and exons among *CsWAK/CsWAKL* genes ranged from 1 to 6 and from 2 to 7, respectively. The promoter regions of 23 *CsWAKs*/*CsWAKLs* possessed diverse cis-regulatory elements that are involved in light, development, environmental stress, and hormone responsiveness. The expression profiles indicated that our candidate genes (*CsWAK1, CsWAK4, CsWAK7, CsWAKL1, and CsWAKL7*) are expressed in leaf tissue. These genes exhibit different expression patterns than their homologs in other plant species. These initial findings are useful resources for further research work on the potential roles of CsWAK/CsWAKL in cellular signalling during development, environmental stress conditions, and hormone treatments.

## 1. Introduction

The cell wall is a physical barrier that maintains the structural integrity of the cell and provides support. This structure is also dynamic enough to allow cell division and expansion during tissue differentiation. Above all, the cell wall, which has a complex network of components such as cellulose, hemicellulose, pectin, and lignin first senses environmental stress and transmits signals that lead to a defensive response [1]. The cell wall component pectin, which is secreted in methyl esterified form and subsequently de-esterified, allows the formation of multiple networks in the cell wall that regulate cell expansion. Pectin is also the target of pathogens [2] and can also be targeted by other noxious plant stress factors [3]. Wall-associated kinases (WAKs) are pectin receptors. WAKs bonded to pectin initiate cell expansion in plant development [4]. Additionally, WAKs bind to small pectic fragments called oligogalacturonides (OGs) induced by pathogen or plant damage and trigger stress response pathways [4]. WAKs contain essential domains including intracellular serine/threonine kinase domain and an extracellular galacturonan-binding or calcium-binding epidermal growth-factor-like (EGF) domain.

In Arabidopsis, five highly similar WAK genes and twenty-two WAK-like genes (WAKLs) were identified based on sequence similarity [5]. Recently, a genome-wide analysis of wall-associated kinase gene families in several species was published. The number of WAK/WAKLs identified in *Oryza sativa* was 125 [6], 115 in *Brachypodium* [7], 39 in *Solanum lycopersicum* [8], 58 in *Gossypium* arboreum, 66 in G. raimondii, 99 in G. hirsutum [9], (Zhang et al. 2021), 68 in *Rosa chinensis* [10], 91 in *Hordeum vulgare* [11], 22 in *Juglans regia* [12], 175 in *Populus trichocarpa* [13], and more than 100 in *Zea mays* [14]. The number of WAK genes in monocot crops seems to be quite large compared with *Arabidopsis* as a dicot, implying possible gene duplication during domestication [15].

Studies have indicated that WAK genes were expressed in expanding cells [16,17,18], pathogen-infected cells [16,17,18], and tissues growing under heavy metal (such as aluminium) stress [19,20,21,22,23,24]. In addition, phenotypes of several WAK mutants demonstrated that WAKs are needed for cell expansion. Antisense expression of WAK4 inhibited cell elongation [25]. Arabidopsis WAK1 and WAK2 are shown to regulate cell wall expansion [23]. Mutants of *wak2* have shorter root hairs suggesting a role of WAKs in cell expansion [26]. RNAi-mediated silencing of *OsiWAK1* in *Oryza sativa* impaired root development [27]. Silenced rice wall-associated receptor-like kinase (OsDEES1) showed fertility deficiency [28].

Evidence that WAK genes play a role in pathogen resistance comes from several studies [29]. Arabidopsis gene *WAKL22* confers resistance to *Fusarium* races [30]. Over-expression of WAK1 in Arabidopsis has contributed to increased resistance to Botrytis [31]. Similarly, over-expression of *OsWAK1* has provided increased resistance to rice blast disease [32]. Silencing tomato WAK1 reduced resistance to a tomato bacterial pathogen [16]. WAKs in *Zea mays* conferred quantitative resistance to maize head smut [14] and northern corn leaf blight [33]. In barley, spot blotch caused by a pathogen *Bipolaris sorokiniana* is one of the significant diseases, which causes 30% yield losses [34]. The *rcs5* quantitative trait locus imparts recessive resistance against the disease spot blotch. An ~234 kb genomic region in this QTL contains four WAK genes, designated *HvWak2, Sbs1, Sbs2* (susceptibility to *B. sorokiniana* 1&2), and *HvWak5* [35].

*Cannabis sativa* L. is the only species of the genus *Cannabis* belonging to the family Cannabaceae which has eight genetically similar genera (*Cannabis, Humulus, Aphananther, Celtic, Gironniera, Lozanella, Pteroceltis,* and *Trema*) [36]. Commercial *Cannabis sativa* L. strains can be separated into two groups of varieties, industrial (hemp) and pharmaceutical (marijuana). A major commercial use of hemp is for cannabidiol (CBD) extraction from unfertilized female flowers (it can be up to 30% of dried flower weight); the value of CBD for various uses (medical, cosmetic, dietary supplement, etc.) in the United States alone was over $4.7 billion (2020 data) [37]. Thus, hemp is certainly a high value crop. In addition to extraction of CBD, other hemp strains are grown commercially for seed production; seeds of hemp are used as nutrients (oil), and the fiber present in other types of hemp is used in the textile industry. The inner hard tissues are used as raw materials in the pulp-making and automotive industry and as insulation material in buildings. Hemp is also an environmentally friendly plant used for phytoremediation in polluted soils [38]. Drug-type cannabis (marijuana) contains up to 30% THC (tetrahydrocannabinol) by dry weight, unlike hemp. Therefore, further research on this versatile herb cultivated in many parts of the world has gained importance.

In this study, genome-wide analysis of the WAK protein family in *Cannabis sativa* L., including the number of family members, protein conserved domains, phylogenetic relationship, subcellular localization, chromosome localization, gene duplication events, gene structural features, and cis-elements, are examined. Furthermore, the expression patterns of some *WAK/WAKL* genes were investigated in roots, stems, leaves, and flowers as well as during flower development to reveal the *WAK/WAKL* gene functions in different organs and flower development.

## 2. Results

### 2.1. Identification and Phylogenetic Analysis of CsWAK/CsWAKL Gene Family in Cannabis

The Arabidopsis WAK and WAKL protein sequences were searched using BLASTp to detect candidate WAK/WAKL proteins in *C. sativa.* A total of 23 *WAK* and 30 *WAKL* protein-encoding genes were identified and numbered in order from CsWAK1 to CsWAK23 and from CsWAKL1 to CsWAKL30 (Appendix A). A total of 53 CsWAK/CsWAKL proteins have been listed according to gene ID and alignment protein sequences in Table 1. All 23 CsWAKs contain characteristic domains, namely GUB-WAK binding, EGF or EGF-Ca, and Pkinase, while CsWAKLs contain pkinase but lack at least one of the other domains (Figure 1). In addition, most of the putative CsWAK/CsWAKL proteins contain a transmembrane domain (TM) and signal peptide (SP) domain. However, CsWAK7, CsWAKL6, and CsWAKL23 lack a TM domain, and CsWAK21 lacks an SP domain. Signal peptide and Gub-wak were located at the N- terminal, while pkinase was at the C-terminal. EGF/ EGF-Ca were located near the transmembrane domain.

A neighbour-joining (NJ) phylogenetic tree was constructed using 53 CsWAK/CsWAKL protein sequences (Figure 2). To reveal evolutionary relationships with 53 CsWAKs/CsWAKLs, some WAKs of Arabidopsis, rice, cotton, and maize that were involved in cell elongation, development, and pathogen resistance were also placed on the phylogenetic tree. All members of the WAK/WAKL family were grouped into four main groups (Group I-IV). The largest group was Group I, which contained 33 family members and was subdivided into 4 sub-groups namely (Group Ia Ib, Ic, and Id). Interestingly, all 5 AtWAK proteins were found in Sub-group Ia and close to 3 GhWAK, followed by 7 CsWAK members, suggesting the closeness of these WAK proteins. Two GhWAK and CsWAKL protein members were found in Sub-group Ic. Group II consisted of 9 CsWAKL protein members. In the second-largest named Group III, 14 CsWAK/CsWAKL and 2 GhWAK proteins were found. Group IV consisted of 4 of the different genome protein sequences, namely 5 OsWAK, 8 CsWAKL, 1 GhWAK, and 1 ZmWAK-RLK1. While 5 of the 6 OsWAK proteins were in Group IV, OsWAK11 was found in the simplicifolious group. ZmWAK-RLK1 and GhWAK5 were found to be clustered in Group IV.

### 2.2. Physiochemical Property and Subcellular Localization Analysis of CsWAKs/CsWAKLs Proteins

The various characteristic properties of the CsWAKs/CsWAKLs proteins are summarized in Table 1. The length of the amino acid sequence of cannabis WAK/WAKL proteins varied from 582 (CsWAKL26) to 983 (CsWAK1) residues, with an average length of 743 aa. The molecular mass ranged from 65.63 (CsWAKL26) to 108.78 (CsWAK1) kDa, with an average molecular mass of 82,92 kDa. The isoelectric point (pI) also varied from 5.08 to 8.96. Additionally, 41 out of the 53 CsWAK/CsWAKL were acidic (pI < 7), and the remaining proteins were basic. The hydrophilic indices of the CsWAKs/CsWAKLs were all less than 0 (−0.109—0.391 GRAVY value), suggesting that they are hydrophilic. Four out of the 53 CsWAK/CsWAKL proteins were classified as unstable proteins. The aliphatic index of all CsWAK/CsWAKL proteins indices ranged from 94.67 to 77.63, which may be regarded as stable under higher temperatures in vivo. All but 3 of the 53 proteins showed an instability index of less than 40 and are probably stable in the test tube.

The subcellular location predicted by WoLF PSORT [39] revealed that CsWAKs/CsWAKLs were localized in the nucleus, cytoplasm, mitochondrion, vacuole, cytoskeleton-nucleus, chloroplast, endoplasmic reticulum, Golgi body, plasma membrane, Golgi-plasma, peroxisome, and the extracellular membrane. However, most CsWAKs/CsWAKLs were primarily localized in the plasma membrane (Figure 3). Peroxisome and cytoskeleton–nucleus contain the least amount of CsWAKs/CsWAKLs.

### 2.3. Chromosomal Location, Synteny, and Selective Pressure of CsWAK/CsWAKL Gene Family

The chromosomal distribution results of 23 *CsWAK/CsWAKL* genes were shown to be unevenly distributed among six chromosomes (Chr) in the cannabis genome (Figure 4). Chr 7 contained the maximum numbers of *CsWAK/CsWAKL* (n = 12), followed by Chr 4 (n = 4), Chr1 (n = 3), and Chr9 (n = 2). While Chr 2 and Chr 8 had the least number, with one *CsWAK/CsWAKL,* Chr 3, 5, 6, and 10 did not contain any *CsWAK/CsWAKL* genes. A total of 30 *CsWAK/CsWAKL* were located on unnumbered Chr. *CsWAKL3/CsWAKL4* on Chr 4 and *CsWAKL7/CsWAK19* on Chr 7 are suggested to have undergone duplication events. According to the above findings, the imbalanced distribution and duplications may have aided in the expansion of the *CsWAK/CsWAKL* gene family. Additionally, to investigate the selective pressure, we calculated the Ka/Ks ratios of 18 linked gene pairs (Table 2). All but one of the Ka/Ks value was less than 1, with CsWAK14/CsWAK5 having the maximum value of 2.268982 and CsWAK19/CsWAKL7 having the minimum value of 0. When 0 < Ka/Ks < 1, this represents purifying selection; Ka/Ks = 1 represents the neutral selection, and Ka/Ks > 1 represents the positive selection [40]. The Ka/Ks values were less than 1, indicating that purification selection kept proteins unchanged. In addition, only one gene each in the Arabidopsis thaliana (CsWAKL14) and Oryza sativa (CsWAK4) genomes and two genes (CsWAKL3 and CsWAKL14) in the *Solanum lycopersicum* genome were found to have a syntenic relationship (Figure 5).

### 2.4. Conserved Motif and Gene Structure of CsWAKs/CsWAKLs

In order to understand the functional diversity and motif distribution among the CsWAK/CsWAKL gene family, the full-length protein sequences were analyzed using MEME (Figure 6b). There were a total of 10 motifs among CsWAK/CsWAKL proteins, and 19 CsWAKs and 9 CsWAKLs contained all ten motifs. Motifs 1 and 6 were found in all CsWAK/CsWAKL proteins. With the exception of CsWAK21 and CsWAKL24, all other proteins contained motifs 3 and 4. Interestingly, all but 1 of the 12 CsWAKLs containing 9 conserved motifs were found to have similar intron–exon composition and were also located on similar branches. Gene cluster (CsWAKL30, CsWAKL11, and CsWAKL21), (CsWAKL16, CsWAKL5), and (CsWAKL19, CsWAKL20, CsWAKL29, CsWAKL10, CsWAKL9, and CsWAKL22) (Figure 6a–c) all contained 2 introns and 3 exons with 9 conserved motifs.

The number of introns and exons among *CsWAK/CsWAKL* genes ranged from 1 to 6 and from 2 to 7, respectively. The 5′ UTR region in *CsWAKL13* and *CsWAKL4* genes were the longest, and, interestingly, they contained 3 introns, 4 exons, and 5 motifs. The highest number of introns and exons (6–7) were found in CsWAK21, while the least number of introns and exons (1–2) were found in CsWAKL7. Genes with similar intron and exon numbers were mostly found to be clustered together.

### 2.5. Cis-Acting Elements Analysis of Promoters of CsWAK/CsWAKL Genes

To investigate the potential regulatory mechanisms of *CsWAK/CsWAKL* genes, the cis-acting elements in the upstream promoter regions of 53 *CsWAK/CsWAKL* genes were examined (Figure 7). A total of 23 *CsWAK/CsWAKL* gene promoters included four types of cis-acting elements, namely light-responsive elements, hormone-responsive elements, environmental-stress-related elements, and development-related elements.

In addition, 25 light-responsive elements were observed in the putative promoter regions. Numerous Box 4 and G-Box cis-acting elements were especially found in 21 *CsWAK/CsWAKL* gene promoters.

Considering hormone responsive elements, a number of abscisic-acid-responsive cis-elements, including ABRE, ABRE3a, and ABRE4, were found within 13 *CsWAK/CsWAKL* gene promoter regions. There were 21 *CsWAK/CsWAKL* genes that possessed at least one cis-acting element involved in the ethylene-responsive element (ERE). Especially, *CsWAK5* and *CsWAKL9* had the largest number of cis-acting elements. At least one MeJA-responsive element (TGACG-motif, TATC-box, and CGTCA-motif) were found in 14 *CsWAK/CsWAKL* gene promoters. The auxin-responsive (AuxRR-core, TGA-box, and TGA-element), gibberellin-responsive element (GARE-motif), and salicylic-acid-responsive (TCA, TCA-element) were observed in several *CsWAK/CsWAKL* gene promotors. Phytohormones, especially auxin and gibberellins, play roles in cell elongation. The abscisic acid, MeJA, and salicylic acid mainly respond to several stress factors and may provide cell signal transduction. The ethylene regulates both growth and senescence. The high expression level of *CsWAK/CsWAKL* genes having cis-acting elements might be induced after these hormone treatments.

Eight types of environmental-stress-related cis-acting elements were discovered in these promotor regions. The following are their names and potential functions: LTR (low-temperature responsiveness), MBS (MYB binding site involved in drought-inducibility), TC-rich repeats (defense and stress responsiveness), ARE (anaerobic induction), box S (elicitation, wounding, and pathogen responsiveness), DRE (involved in dehydration, low-temp, and salt stresses), W box (wounding and pathogen responses), and WUN-motif (wound-responsive elements). There were 20 *CsWAK* and 15 *CsWAKL* gene promoters that had ARE and WUN motifs, respectively. CsWAK5 had the largest number of WUN-motif.

Finally, development-related elements, AAGAA-motif (involved in the endosperm-specific negative expression), AC-I (involved in xylem-specific expression), O2-site (involved in zein metabolism), as-1 (root-specific expression), CAT-box (meristem expression), CCAAT-box (meristem expression), circadian (circadian control), GCN4_motif (endosperm expression), RY-element (seed-specific regulation), HD-Zip-1 (differentiation of the palisade mesophyll cells), and HD-Zip 3 (protein binding site), were also found in the putative promoter regions of 23*CsWAK/CsWAKL* genes. Only *CaWAKL11* promotor had a seed-specific regulation element (RY-element). Twelve of the 23 *CsWAK/CsWAKL* genes had root-specific elements (as-1) of which *CsWAK3* had the largest number of RY elements.

### 2.6. Expression Patterns of the Some CsWAKs/CsWAKLs Genes in Different Tissues

To analyze the function of CsWAKs and CsWAKLs, we selected some CsWAKs/CsWAKLs candidate genes to identify their expression pattern in different cannabis tissues. The homolog of CsWAKL1 and CsWAKL7, *Juglans regia* WAK2 and WAKL13, respectively, are mainly expressed in female flowers. JrWAK9, the homolog of CsWAK1, is involved in pathogen response [12]. *AtWAKL22* gene, the homolog of *CsWAK*7, is responsible for dominant resistance against several Fusarium races [30]. The cotton WAK18 (CsWAK4) and WAK29 (CsWAK7) are specifically expressed in flowers [41,42]. Therefore, we chose *CsWAK1, CsWAK4, CsWAK7, CsWAKL1*, and *CsWAKL7* as our candidate genes. Figure 8 shows that these five genes were mainly expressed in leaf tissue. The expression levels of *CsWAK1, CsWAK4, CsWAK7,* and *CsWAKL7* were more than 3-fold greater in the leaf as compared with the root. These genes exhibited different expression patterns than their homologs in other plant species. This finding could provide a basis for identifying new possible roles the CsWAK and CsWAKL gene products play in cannabis development and response to environmental stresses.

### 2.7. Identification of Protein–Protein Interaction and miRNA Targets 

Protein–protein interaction was performed using WAK/WAKL orthologous in *Arabidopsis thaliana*. In Figure 9, the interaction network showed 8 Arabidopsis WAK/WAKL proteins associated with 43 CsWAK/CsWAKL proteins (Appendix A). A total of 24 CsWAK/CsWAKL proteins matched with WAK2, followed by 7 and 4 with WAK5 and WAK1, respectively. WAKL8, WAKL9, and WAKL22 were associated with 2 CsWAK/CsWAKL each. Finally, only CsWAKL matched with AT1G18390 and AT1G69730 (AtWAKL9). Out of the 53 CsWAK/CsWAKL, 10 proteins corresponding to *Arabidopsis thaliana* orthologous did not show in the network interaction because disconnected nodes in this network were hidden. From the results, the number of nodes and edges were 23 and 25, respectively. The PPI enrichment value obtained was 0.000105, demonstrating that the network contained substantially more interactions than was anticipated. An enrichment of such indicates that the proteins are at least partially biologically related in nature. WAK1 was found to be highly connected among the interacted proteins.

MicroRNAs (miRNAs) play important roles in gene regulation. In this present study, it was investigated whether csa-miRNAs have target regions in CsWAK/CsWAKL transcripts to reveal miRNAs that can regulate *CsWAK/CsWAKL* genes. For this purpose, CsWAK/CsWAKL transcripts were searched using 9 csa-miRNAs belonging to 6 independent families to detect the target sites with csa-miRNAs. These miRNA sequences were retrieved from Das et al. (2015) [43] and Hasan et al. (2016) [44]. Finally, two miRNAs (csa-miR172 and csa-miR5021) targeted with CsWAKs/CsWAKL transcripts. csa-miR5021 was found to be the common miRNA and targeted 5 CsWAK/CsWAKL transcripts. csa-miR5021 interacted with CsWAK6, CsWAK9, CsWAKL4, and CsWAKL5 via cleavage, and interacted with CsWAKL13 and CsWAKL14 via translational inhibition. However, csa-miR172 targeted only CsWAK5 via cleavage (Appendix A).

## 3. Discussion

*Cannabis sativa* L. is both an ecologically and economically important crop that has been used to produce textiles, paper, building materials, food, and medicine. The first genomic draft of cultivated variety in *C. sativa* was published in 2011 [45], following genomic sequencing of wild-type varieties [46]. With the publication of genome sequencing, genome-wide identification of key gene families in this species has been carried out in recent years [47,48,49,50,51,52,53,54]. WAK gene family members trigger cell elongation during plant development by binding to cross-linked pectin in the cell wall and provide a response to pathogen or environmental stress factors by binding to OGs (oligo-galacturonides) formed by pathogens or mechanical damage [55]. Thus, in this study, a comprehensive genome-wide analysis of wall-associated kinases in *C. sativa* and their expression levels in different tissues and flower development was performed. A total of 23 CsWAK and 30 CsWAKL protein family members were identified. Previous studies showed that the number of WAK/WAKL genes differed according to species, regardless of genome size—for example, 26 in Arabidopsis [22], 29 in cotton [41] and 29 in tomato [8], 125 in rice [56], 91 in barley [41], 68 in rose [10], 115 in Brachypodium [7], and more than 100 in maize [14].

To clarify the evolutionary relationship among the members of WAK gene family, a phylogenetic tree was constructed using the WAK/WAKL protein sequences from *Arabidopsis thaliana*, *Oryza sativa*, *Zea mays,* and *Gossypium hirsutum.* We compiled a total of eight pathogen-defense-response-related WAK proteins including OsiWAK112, OsWAK91, OsWAK92, OsWAK14 from Oryza sativa, GhWAK5, GhWAK9, GhWAK26 from *Gossypium hirsutum*, and ZmWAK-RLK1 from *Zea mays*. Additionally, WAK proteins differentially expressed in tissues (GhWAK3, GhWAK4, and OsiWAK1) and defense response against excess copper (OsWAK11) were included. Phylogenetic analysis has shown that all CsWAKs/CsWAKLs were classified into four main groups (Figure 2). Arabidopsis WAKs were grouped together into Subgroup 1. The fact that all Arabidopsis WAKs were in the same subgroup and that other OsWAKs and GhWAKs were generally clustered close to each other was consistent with the findings of previous studies in cotton [40], in *Junglas regia* [12], in apple [57], and in barley [11]. However, the exception is that OsiWAK11 was not close to the others. The species–specific grouping of WAKs by phylogenetic analyses might be explained by the independent evolution of WAKs in each species. Moreover, the phylogenetic groups indicated that WAKs/WAKs have a high diversity of across species.

According to chromosomal localization, 23 *CSWAK/CsWAKL* genes were unevenly distributed on six of the cannabis chromosomes, but most of them (12 ***CSWAK/CsWAKL***) were located on Chr 7 (Figure 4). Furthermore, two tandem duplications occurred on Chr 4 and Chr 7, including four genes (CsWAKL3-CsWAKL4 and CsWAKL7-CsWAK19). Additionally, the flanking region of *CsWAK1* and *CsWAKL24* genes contain reverse transcriptase-like and zinc-binding in reverse transcriptase sequences, respectively. This finding is consistent with a previous study [58] concerning Arabidopsis *WAK1, WAK2, WAK4, WAK5, WAKL11,* and *WAKL16* genes carrying Hopscotch (Copia like retrotransposon). Kumar et al. [59] underlined that retrotransposons may carry all or part of the gene nearby, causing them to expand and spread throughout the genome. Similarly, the increase in the number of members of the WAK gene family by tandem or segmental duplications and retrotransposon and the acquisition of new functions in duplicated genes have been demonstrated in various studies [7,8,60].

To gain a better understanding of the CsWAK/CsWAKL gene family’s evolutionary relationship, we calculated the Ka/Ks ratios of 18 linked gene pairs (Table 2). We found that all but one of the Ka/Ks values was less than 1. This indicated that the pattern of evolution was mainly by purifying selection except for one with positive selection, hence, critical in sustaining the number of gene family members.

Classification of the members of a gene family is an essential step in exploring their structural and functional characteristics as well as evolutionary relationships [51]. In this respect, the protein motifs and their distribution were investigated to reveal their functional diversity. A total of 10 motifs, ranging from 17 to 50 amino acids length were found in 53 CsWAK/CsWAKLs. Half of the 10 predicted motifs were associated with WAK protein domains. Four motifs (motifs 1, 2, 5, and 6) corresponded to pkinase domain and one (motif 7) to the EGF domain. CsWAKs/CsWAKLs belonging to the same group/subgroup in the phylogenetic tree (Figure 6a) showed highly similar motif distribution. Also, the fact that each group/subgroup of CsWAK/CsWAKLs had different motifs from each other may be attributed to the diversity of their functional roles.

The presence of cis-elements in the promoter sequences suggests regulation of the transcription of these genes that may be responsible for effects on development and response to hormones and stress. Therefore, cis-acting elements were analyzed in this study. A total of 23 *CsWAK/CsWAKL* gene promoters contained light-response elements, especially *CsWAKL11* and *CsWAKL9* which had the most; additionally, 21 *CsWAK/CsWAKL* had Box 4 cis-elements. Eleven *CsWAK/CsWAKL* promoters had the cis-element associated with endosperm-specific negative expression. *CsWAK3* contained only the cis-elements (as-1) for root-specific expression. The seed-specific regulation (RY-element) element was found only in the promoter of the *CsWAKL11* gene. Therefore, the development-related cis-element analysis indicates tissue-specific expression of certain *CsWAK/CsWAKL* genes. The promoters of *CsWAK7, CsWAK9, CsWAKL12,* and *CsWAKL10* genes had circadian-rhythm-responsive elements. This finding was consistent with previous studies pointing to the regulation of some members of receptor-like kinases by the circadian clock [61]. Considering hormone response, abscisic acid, Methyl jasmonic acid, auxin, gibberellin, salicylic acid, and ethylene hormone-responsive cis-elements were found in 13, 15, 12, 1, 19, and 21 *CsWAK/CsWAKL* genes, respectively. This means that the expression of these genes might be induced by treatment with the respective hormones. In addition, these *CsWAKs/CsWAKLs* that have hormone-related elements may play an important role in cell elongation and stress response, as hormones have important effects on plant growth, response to stress, and adaptation to adverse environmental conditions. Likewise, a previous study showed that the expression level of 13 and 10 GhWAK genes containing the gibberellin- and the auxin-responsiveness element, respectively, were increased after gibberellin and auxin treatment [41].

In this study, many genes closely located on the same chromosome and in the same subgroup of the phylogenetic tree constructed using 10 conserved motifs had similar exon–intron distribution, for example, *CsWAKL2, CsWAKL3,* and *CsWAKL4* in Chr 4; *CsWAK*5, *CsWAK6, CsWAKL9,* and *CsWAKL10* on Chr 7; and *CsWAKL13-CsWAKL14* in Chr 9. These genes differed only in the length of introns and upstream regions. Similarly, Dou et al. (2021) [41] observed that genes with high sequence similarity in cotton are located close to each other on the same chromosome and reported that this would prevent loss of function during evolution.

Gene expression pattern analysis could predict systematic gene function [62]. Known WAK/WAKL genes in other plant species could give us an effective way to select candidate WAK/WAKL genes in cannabis [12,41]. Cannabis is one of the most important commercial crops and has a long history during the development of human civilization [63]. Cannabis flower tissue, especially trichomes, has a central place in industry and academic research [64]. We used homolog analysis to select four CsWAK/CsWAKL genes (predicted in flower development) and one gene (predicted in pathogen response). Interestingly, these four CsWAK/CsWAKL genes (*CsWAK1, CsWAK4, CsWAK7*, and *CsWAKL7*) were mainly expressed in leaf tissue, not flowers. This could give us new thoughts about the function of these genes. Therefore, our work suggests that further research to explore the function of these gene products in leaves is warranted.

Protein–protein interaction analysis was conducted to predict the potential function of CsWAK/CsWAKL proteins. WAK1 was seen to be the most interactive protein (Figure 9). This also confirms the strong interaction between Gly-rich extracellular protein AtGRP-3 [65] and a cytoplasmic type 2C protein phosphatase KAPP [66] with WAK1 to generate a 500-kD protein complex (Figure 9). GRP-3 expression is upregulated in response to ABA, salicylic acid, and ethylene but downregulated in response to desiccation. Furthermore, KAPP plays a role as a negative regulator of the CLV1 signaling in plant development. In addition, WAK1 protein is required during the plant’s response to pathogen infection and in plant defense against heavy-metal toxicity [5]. WAK1 had four CsWAK/CsWAKL proteins (Appendix A) corresponding to it, suggesting that four CsWAK/CsWAKLs may share in the fundamental roles of WAK1 and its complex with GRP-3 and KAPP. According to Wagner and Kohorn (2001) [23], WAK2 plays an important role in cell elongation. WAK2 corresponded to 24 CsWAK/CsWAKL proteins. Thus, these CsWAKs/CsWAKLs participate in cell elongation. Furthermore, interactions between RFO1(AtWAKL22) corresponding to CsWAKL12, CsWAKL18, and RLP3 could also be associated with their similarity in conferring disease resistance to a broad spectrum of *Fusarium* races, indicating that CsWAKL12 and CsWAKL18 have roles in response to abiotic stress. AT1G16260 matched with CsWAK22 and CsWAK9 functions as a signaling receptor of extracellular matrix components, and it interacted only with AT4G01860 and AT5G58412.

miRNAs are small, non-coding RNA molecules. They regulate gene expression at transcriptional and translational levels and complex biological processes [67]. Das et al. (2015) mined miRNAs in silico for the first time in *C. sativa*, followed by Hasan (2015) [44]. A total of 18 conserved miRNAs belonging to nine families were found by homology-based computational approach in Transcript Sequence Assemblies (TSA) and ESTs data in *C. sativa* [43,44]. The number of members in each of the miRNA family varied from one to seven. A total of 18 csa-miRNAs targeted 80 genes in the Arabidopsis genome, including transcription factors, transporters, kinase, and other enzymes as well as signalling and stress-responsive proteins. In the present study, csa-miR5021 and csa-miR172 were found to be related with regulation of three *CsWAK* genes (*CsWAK5, CsWAK6, and CsWAK9*) and four *CsWAKL* genes (*CsWAKL4, CsWAKL5, CsWAKL13, and CsWAKL14*) via the process of cleavage or translation (Appendix A). csa-miR172 targeted CsWAK5 transcript. It has been reported that the csa-miR172 family has seven members and csa-miR172-mediated gene regulation in C. sativa plays dominant roles. Moreover, csa-miR172 were found to act on transcription factors that regulate flowering time in *C. sativa* (Das et al. 2015) and also promoted flowering and adult leaf traits in Arabidopsis [68]. In addition, Das et. al. (2015) [43] concluded that the genes targeted by csa-miR5021 and csa-miR6034 were involved in ribosomal proteins, calcium-binding EF hand family protein, global transcription factor, disease resistance protein, leucine-rich receptor-like protein kinase, and so on. Thus, these csa-miRNAs participate in growth, development, and stress responses in *C. sativa*. Further studies are needed to elucidate CsWAK/CsWAKLs’ roles in plant growth and stress response through miRNA-mediated silencing of CsWAK/CsWAKLs.

## 4. Materials and Methods

### 4.1. Identification of WAK/WAKL Genes

To identify the WAK/WAKL genes, protein sequences of *AtWAK/AtWAKL* genes from (https://www.ncbi.nlm.nih.gov/, accessed on 16 March 2022) were used as query sequences and blasted (BlastP) against the cannabis genome database with an E-value of 1e-5. Redundant sequences were removed. The final WAK/WAKL gene family members were determined by analyzing the protein domains of all candidate genes. The candidate proteins containing all 3 domains (wall-associated receptor kinase galacturonan-binding domain, calcium-binding EGF or EGF domains, and protein kinase domain/protein tyrosine kinase domains) were considered as WAK proteins using Pfam 35.0 online software (http://pfam.xfam.org/, accessed on 18 March 2022) [69] and NCBI-CDD (https://www.ncbi.nlm.nih.gov/Structure/cdd/wrpsb.cgi? accessed on 18 March 2022) [70]. The proteins that did not contain GUB WAK or EGF/EGF Ca domains were recognized as WAKL proteins. Lastly, the signal peptide and transmembrane domain were checked with HMMER online software 2.41.2 (https://www.ebi.ac.uk/Tools/hmmer/search/phmmer, accessed on 24 March 2022).

### 4.2. Physicochemical Properties, Subcellular Prediction, and Conserved Motif Analysis of WAKs/WAKLs

The physicochemical properties of WAKs/WAKLs were predicted using Expasy (https://web.expasy.org/protparam/, accessed on 14 April 2022) [71], an online website. Subcellular localization of all identified WAK/WAKL proteins was detected using the online site WoLF PSORT (https://wolfpsort.hgc.jp/, accessed on 1 April 2022) (Horton). HeatMap was constructed using TBtools software [72]. MEME 5.4.1 (https://meme-suite.org/meme/tools/meme, accessed on 27 March 2022) was used to analyze the conserved motifs of WAK/WAKL proteins, using default settings except for the number of motifs, 10; minimum and maximum motif widths, 6–50 [73].

### 4.3. Phylogenetic Tree Construction and Gene Structure Analysis Identification of WAK/WAKL Genes

The neighbor-joining approach was used to infer the evolutionary history [74]. A total of 1000 replicates yielded a bootstrap consensus tree [75]. The p-distance method was used to calculate the evolutionary distances [76]. MEGA 11 was used to undertake evolutionary analysis [77]. ClustalW was used for multi sequence alignment. The gene structure analysis was performed using the online Gene Structure Display Server (http://gsds.gao-lab.org/, accessed on 31 March 2022).

### 4.4. Chromosome Localization, Synteny Analysis, and Ka/Ks Calculation

The structure of the WAK/WAKLs and their localization on the chromosomes were visualized and embellished using TBtools software [72]. Synteny analysis was accomplished using genome files of both *A. thaliana* and *C. sativa* species. TBtools software was used to synteny analysis and calculate the Ka/Ks ratio [72].

### 4.5. Prediction of Cis-Acting Elements

The 2000 bp upstream promoter sequences of *WAK/WAKLs* were manually extracted from NCBI website (https://www.ncbi.nlm.nih.gov/, accessed on 7 April 2022)). Cis-acting elements were predicted using PlantCARE (http://bioinformatics.psb.ugent.be/webtools/plantcare/html/, accessed on 9 April 2022), an online software. (lescot) A draft cis-acting element sheet was obtained from Lescot et. al. (2002) [78] and the values of our obtained cis-acting elements were manually inputted. The cis-acting elements were categorized into four major groups: light-responsive elements, hormone-responsive elements, environmental-stress-elements, and development-related elements. HeatMaps were drawn using TBtools software V1.098769 (Chengjie Chen, Guangdong, China) [72].

### 4.6. Plant Material

The hemp variety, ‘Wife’, was used in this study. Cuttings of this variety were taken from 5-month-old ‘Wife’ mother plants and were rooted in an EZ-Cloner Classic Chamber™ (Sacramento, CA, USA) under aeroponic conditions. Cuttings were treated with Hormodin powder and placed in rockwool cubes soaked with 20 mL/L Clonex Nutrient Solution (Growth Technologies Ltd., Taunton, UK). Cuttings were allowed to root for 3 weeks before transplant. Rooted cuttings were potted into a 3-gallon size container with Pro Mix HP and were given liquid Botanicare™ (Vancouver, WA, USA) fertilizer two times per week. After 8 weeks, they were transplanted into a 10-gallon container. Vegetative growth continued for an additional week. During vegetative growth, plants were grown under 18 h light/6 h darkness daily cycle (vegetative growth cycle). During flowering growth, plants were grown under 12 h light/12 h darkness to initiate floral development. The plants were maintained in this environment for 7 weeks. During the vegetative phase, liquid feed was supplied in the form of Jack’s Nutrients at 100 ppm N with every watering. During the flowering phase, Jack’s 15-30-15 was fed at 100 ppm N with every watering. Irrigation was supplied by Netafim drip assemblies as needed.

### 4.7. Total RNA Isolation, cDNA Synthesis, and Gene Expression Analysis

The amount of 100 mg of plant tissues (root, leaf, flower, and trichomes) was collected and immediately frozen in liquid nitrogen. The NucleoSpin Plant and Fungi RNA Isolation Kit (Macherey-Nagel) was used for RNA isolation according to the manufacturer’s manual with several modifications. cDNA was synthesized from 2 µg RNA using the iScript Reverse Transcriptase Master Mix (BioRad).

qPCR analysis was performed using Bio-Rad CFX. iTaq Universal Sybr Green Master Mix (Hercules, CA, USA) was used (Bio-Rad). For all qPCR reactions, CsUbiquitin (CsUBQ) was used as the internal reference [79]. Table 3 shows the qPCR primers used for five of the WAK genes mentioned in this report. Primers were designed using PerlPrimer software (v1.1.21) (PerlPrimer for Microsoft Windows, Owen J Marshall, Australia) [80]. Experiments were performed with four biological replications.

### 4.8. Isolation of Cannabis Trichomes

Trichome isolation was performed following the protocol developed by Livingston et al. 2020 [81] with several modifications. Trichomes collected were detected using microscope. The content of trichomes is 100–500 mg which was from 5 g flower samples. Isolation buffer was made fresh during different experiments. We used W5 solution (2 mM MES (pH 5.7) containing 154 mM NaCl, 125 mM CaCl2, and 5 mM KCl) for trichome isolation. Approximately 5 g of fresh inflorescence tissue was used for trichome isolation. After isolation, the tube containing approximately 100–500 mg of trichome tissue enriched with glands was then frozen at −80 °C until subsequent use for RNA isolation.

### 4.9. Identification of Protein-Protein interactions and miRNA Target Sites

The protein–protein interaction analysis was carried out on an online platform STRING (https://string-db.org/, accessed on 4 October 2022) [82]. Orthologous WAK/WAKL protein sequences from *Arabidopsis thaliana* were used in identifying significant protein–protein interactions. Using the parameters and medium confidence (0.400), no more than 10 interactions were utilized to show and hide disconnected nodes in the network.

The pSRNATarget software (https://www.zhaolab.org/psRNATarget/, accessed on 7 October 2022) [83] was used to identify the miRNA target sites in CsWAK/CsWAKL transcripts. The maximum expectation values of 5.0, target accessibility of 25.0, and the value for gap penalty and seed region for the mismatches of 2 parameters were used to perform the analysis.

## 5. Conclusions

In the present study, WAK gene family in *Cannabis sativa* was genome-wide identified and characterized based on the gene structure, phylogenetic relations, chromosomal location, subcellular localizations, physico-chemical properties, cis-acting element in the promoter region, and protein–protein interactions. Fifty-three CsWAK/CsWAKL family members were identified and classified into four main groups which included species–specific subgroups. CsWAK/CsWAKL family members showed structural variations, indicating functional variations. Gene expression analysis showed candidate genes were primarily expressed in leaf tissues and had different expression patterns compared with their homologs in other plant species.

## Figures and Tables

**Figure 1 plants-11-02703-f001:**
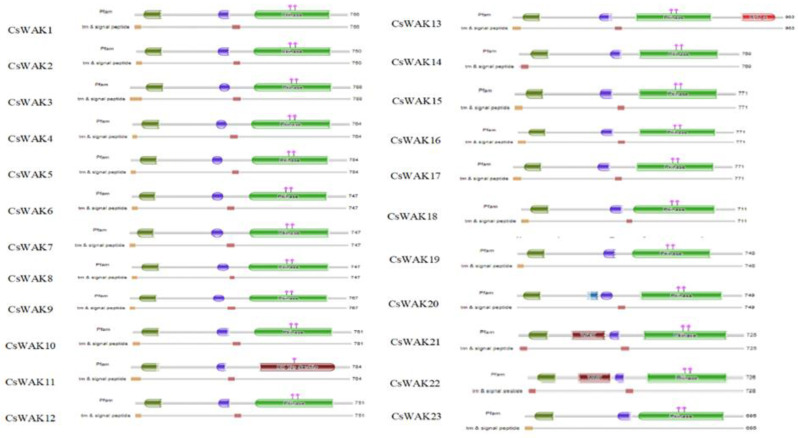
Protein domains of CsWAK/CsWAKL family members. Different domains were represented in green; the red, blue, and dark green colored boxes represented protein kinase and protein tyrosine kinase domains, calcium-binding EGF or calcium-binding EGF-CA domain, and wall-associated receptor kinase galacturonan-binding domain, respectively. The catalytic residues of the CsWAKs were indicated with the pink diamond and vertical poles on pkinase domain. Brown and dark brown boxes showed transmembranes and signal peptides.

**Figure 2 plants-11-02703-f002:**
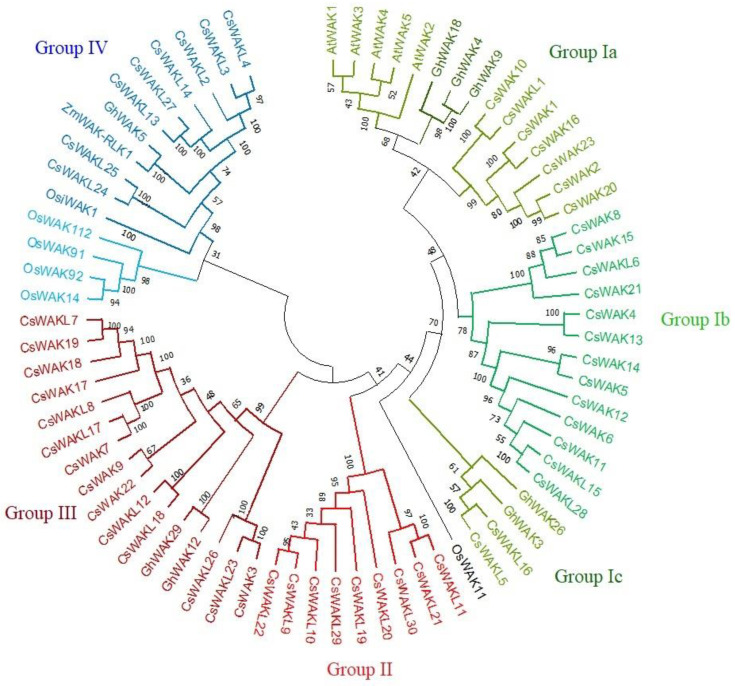
The neighbour-joining phylogenetic tree of WAK/WAKL proteins of *Cannabis sativa*, *Arabidopsis thaliana, Gosyphium hirsutum, Oryza sativa,* and *Zea mays* was conducted using MEGA 11 with a bootstrap of 1000 replicates, based on 95% shared amino acid sites. The tree was divided into four groups: Group I, Group II, Group III, and Group IV containing green, red, maroon, and blue colors, respectively.

**Figure 3 plants-11-02703-f003:**
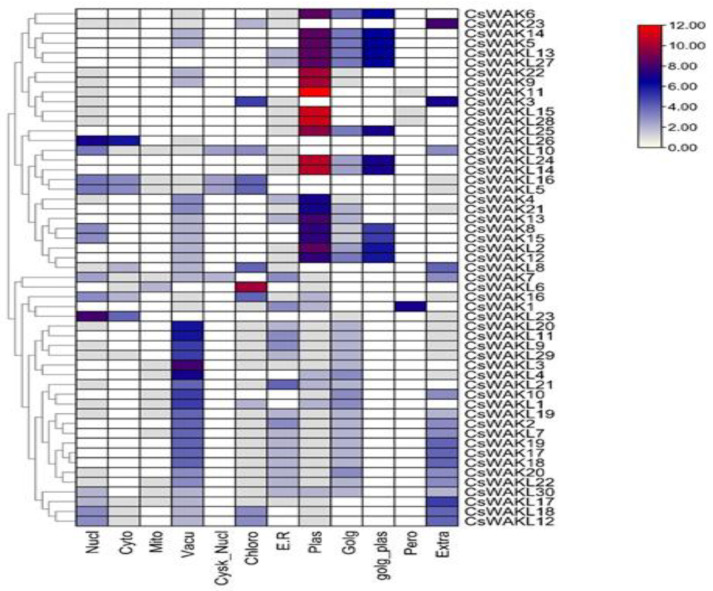
Prediction of subcellular location of *CsWAK*/*CsWAKL* genes through WoLF PSORT online platform.

**Figure 4 plants-11-02703-f004:**
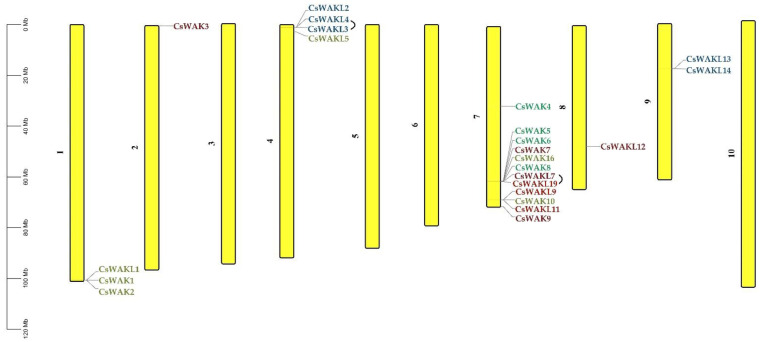
Chromosomal distribution of 23 WAK/WAKL genes in *Cannabis sativa.* Chromosome numbers are shown beside each chromosome. Colors of genes were obtained from the phylogenetic tree.

**Figure 5 plants-11-02703-f005:**
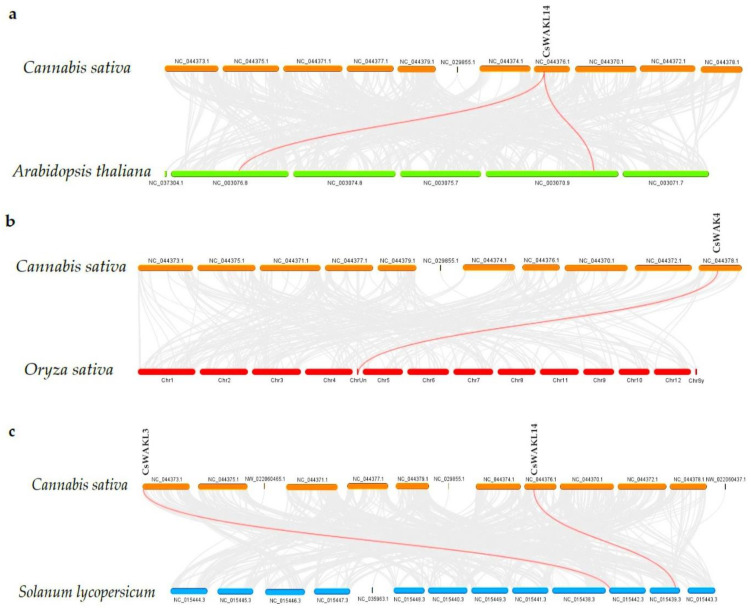
Synteny analysis of WAK/WAKL genes between *Cannabis sativa* and *Arabidopsis thaliana* (**a**); *C. sativa* and *Oryza sativa* (**b**); and *C. sativa* and *Solanum lycopersicum* (**c**). The red lines represent the syntenic WAK/WAKL gene pairs, whiles the grey lines represent the syntenic blocks within the two genomes.

**Figure 6 plants-11-02703-f006:**
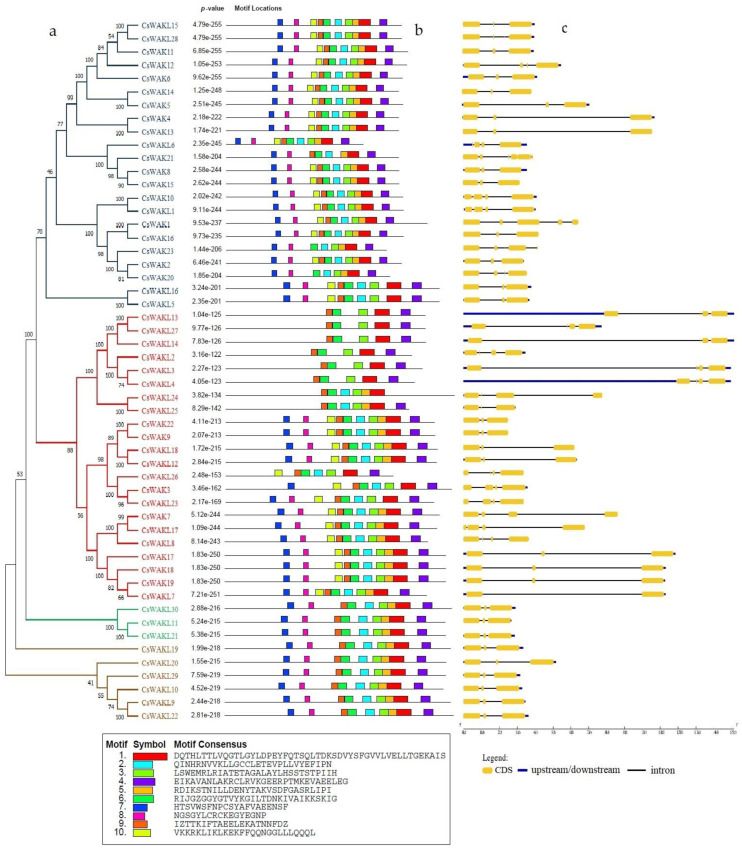
The phylogenetic tree of CsWAK/CsWAKL proteins in *Cannabis sativa* constructed with 1000 bootstrap using the neighbor-joining algorithm (**a**) and the motif sites of the CsWAK/CsWAKL protein sequences (**b**). The height of the block gives the indication of the significance of the site, where taller blocks are more significant. (**c**) Gene structure of *CsWAK*s/*CsWAKLs*. The yellow boxes, blue lines, and black lines represent the CDS regions, upstream/downstream regions, and the introns, respectively.

**Figure 7 plants-11-02703-f007:**
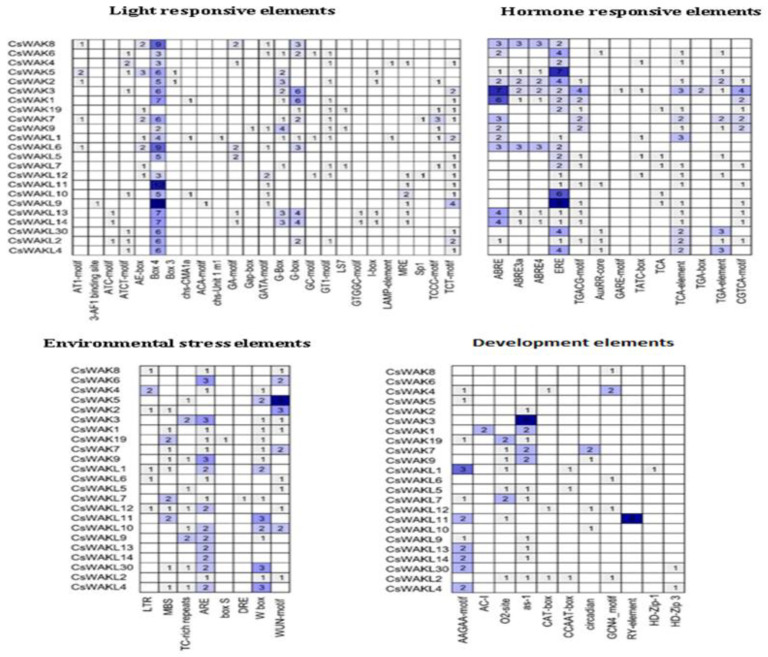
Cis-acting elements in promoter regions of *CsWAK*s/*CsWAKLs* genes. On the basis of functions, they were categorized into four major groups: light-responsive elements, hormone-responsive elements, environmental-stress-related elements, and development-related elements. The numbers in the boxes represent the number of cis-acting elements. The background color changes from light blue to dark blue according to the increasing number of cis elements.

**Figure 8 plants-11-02703-f008:**
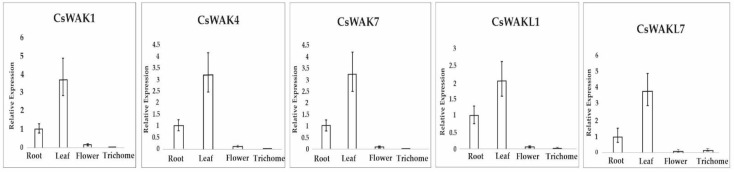
The expression pattern of selected *CsWAK/CsWAKL* genes in root, leaf, flower, and trichome. The ∆∆CT method was used for calculations. X-axis refers to different tissues. Data are presented as means ± SE (n = 3): mean separation between expression in various tissues compared with the level in the root.

**Figure 9 plants-11-02703-f009:**
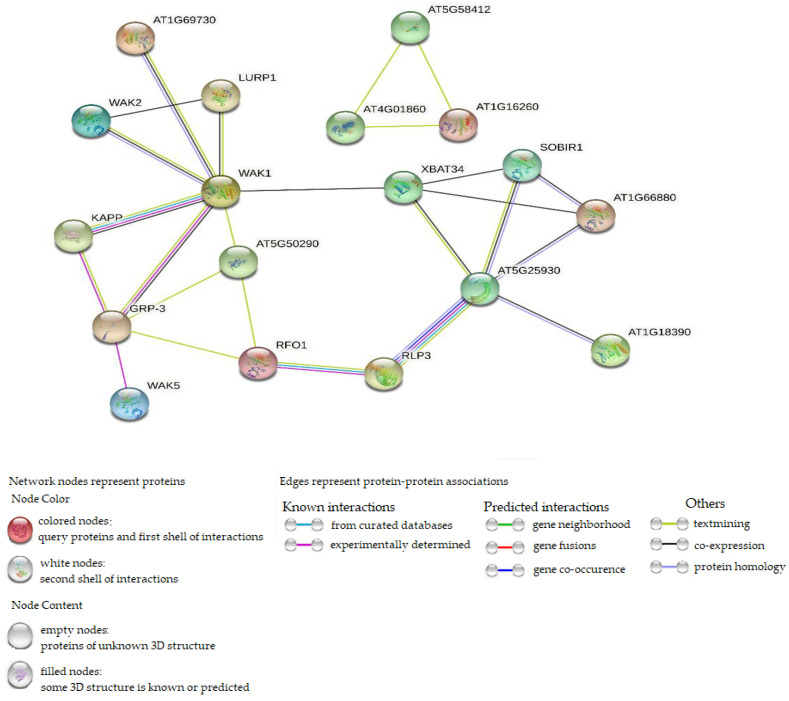
Protein–protein interaction of CsWAK/CsWAKL and Arabidopsis WAK/WAKL orthologous, setting the network edges as evidence.

**Table 1 plants-11-02703-t001:** The characteristics of 53 CsWAK/CsWAKL proteins in *Cannabis sativa*.

Protein ID	Gene Name	Length (aa)	Molecular Weight (kDa)	pI	Grand Average of Hydropathicity (GRAVY)	Aliphatic Index	Instability Index
XP_030487710.1	CsWAK1	983	108.78	6.45	−0.109	94.67	34.45
XP_030492016.1	CsWAK2	761	84.63	6.55	−0.125	91.94	34.58
XP_030503825.1	CsWAK3	784	87.58	5.29	−0.302	80.54	41.03
XP_030480968.1	CsWAK4	747	83.83	6.25	−0.210	89.63	36.68
XP_030481386.1	CsWAK5	767	85.73	6.74	−0.234	86.53	31.08
XP_030479582.1	CsWAK6	764	85.62	8.09	−0.250	86.24	33.42
XP_030479448.1	CsWAK7	749	84.33	5.65	−0.332	79.63	38.18
XP_030481338.1	CsWAK8	750	84.58	5.64	−0.236	78.61	26.68
XP_030479562.1	CsWAK9	726	80.08	5.67	−0.059	90.61	35.50
KAF4346698.1	CsWAK10	766	84.17	5.08	−0.203	88.15	36.29
KAF4398384.1	CsWAK11	788	88.67	8.33	−0.264	85.34	35.12
KAF4376937.1	CsWAK12	784	87.67	7.79	−0.247	86.52	31.11
KAF4391022.1	CsWAK13	747	83.82	8.96	−0.212	88.84	37.39
KAF4376938.1	CsWAK14	747	83.69	6.46	−0.264	84.67	31.97
KAF4376931.1	CsWAK15	751	84.65	5.65	−0.226	79.55	26.74
KAF4346705.1	CsWAK16	769	85.46	6.26	−0.221	89.45	35.87
KAF4379309.1	CsWAK17	771	86.44	5.52	−0.154	84.03	33.17
KAF4376930.1	CsWAK18	771	86.39	5.58	−0.159	84.03	33.75
XP_030480237.1	CsWAK19	771	86.36	5.52	−0.162	83.90	32.68
KAF4353681.1	CsWAK20	711	78.60	6.06	−0.091	91.81	31.83
KAF4398399.1	CsWAK21	748	83.54	5.08	−0.153	82.47	26.33
KAF4377370.1	CsWAK22	725	79.97	5.81	−0.057	90.59	35.02
KAF4346699.1	CsWAK23	695	76.92	5.73	−0.112	90.27	33.03
XP_030492015.1	CsWAKL1	769	84.33	5.15	−0.179	89.08	34.43
XP_030496407.1	CsWAKL2	646	72.77	6.67	−0.274	87.51	45.51
XP_030496276.1	CsWAKL3	682	76.40	5.55	−0.200	87.04	42.61
XP_030496277.1	CsWAKL4	656	74.00	6.92	−0.295	83.95	42.72
XP_030499458.1	CsWAKL5	741	83.02	6.26	−0.266	87.44	33.63
XP_030481339.1	CsWAKL6	594	67.26	5.92	−0.295	80.89	26.93
XP_030480238.1	CsWAKL7	704	78.95	5.62	−0.163	85.53	32.25
KAF4376932.1	CsWAKL8	708	80.00	6.28	−0.284	82.87	34.43
XP_030479527.1	CsWAKL9	787	87.47	7.63	−0.320	80.97	35.08
XP_030480670.1	CsWAKL10	763	84.55	8.27	−0.292	81.73	34.78
XP_030480956.1	CsWAKL11	769	85.81	6.63	−0.356	82.46	30.83
XP_030484049.1	CsWAKL12	732	81.76	6.90	−0.164	87.84	37.74
XP_030508821.1	CsWAKL13	692	77.11	5.28	−0.319	79.31	36.59
XP_030508820.1	CsWAKL14	694	76.93	5.58	−0.269	81.59	36.30
KAF4379299.1	CsWAKL15	761	85.76	7.42	−0.259	86.32	35.33
KAF4355457.1	CsWAKL16	741	83.08	6.18	−0.263	86.77	33.15
KAF4398398.1	CsWAKL17	740	83.41	5.52	−0.307	81.26	38.47
KAF4381619.1	CsWAKL18	735	82.07	6.65	−0.165	88.54	39.53
KAF4365316.1	CsWAKL19	788	87.85	8.10	−0.253	87.42	34.39
KAF4365315.1	CsWAKL20	773	85.62	7.64	−0.334	82.42	32.16
KAF4385963.1	CsWAKL21	771	86.07	6.51	−0.357	82.24	30.78
KAF4385975.1	CsWAKL22	798	88.67	7.63	−0.278	84.25	35.22
KAF4385447.1	CsWAKL23	723	80.87	5.47	−0.391	77.63	37.34
KAF4363501.1	CsWAKL24	794	87.89	8.16	−0.111	91.56	35.43
KAF4346333.1	CsWAKL25	636	69.66	6.30	−0.143	88.58	35.78
KAF4349269.1	CsWAKL26	582	65.63	5.36	−0.388	84.55	38.31
KAF4349872.1	CsWAKL27	692	77.01	5.28	−0.318	79.31	36.59
KAF4379299.1	CsWAKL28	761	85.76	7.42	−0.259	86.32	35.33
KAF4365319.1	CsWAKL29	770	85.42	6.27	−0.286	85.29	32.59
KAF4365318.1	CsWAKL30	791	88.21	6.25	−0.271	88.31	33.10

**Table 2 plants-11-02703-t002:** Selective pressure analysis.

Gene Pair		Ka	Ks	Ka/Ks
CsWAKL15	CsWAKL28	0	0	NaN
CsWAK14	CsWAK5	0.004595	0.002025	2.268982
CsWAK4	CsWAK13	0.004019	0.004055	0.991173
CsWAK8	CsWAK15	0.003109	0.018028	0.172439
CsWAK10	CsWAKL1	0.009062	0.025339	0.357635
CsWAK1	CsWAK16	0.012437	0.042493	0.292677
CsWAK2	CsWAK20	0.00731	0.029908	0.244432
CsWAKL16	CsWAKL5	0.003498	0.036624	0.095504
CsWAKL13	CsWAKL27	6.31E-04	0.006127	0.103065
CsWAKL3	CsWAKL4	0.206354	0.304622	0.67741
CsWAKL24	CsWAKL25	0.025859	0.05379	0.480727
CsWAK22	CsWAK9	0.003592	0.006016	0.597084
CsWAKL18	CsWAKL12	0.005622	0.03376	0.166527
CsWAK3	CsWAKL23	0.01581	0.022952	0.688858
CsWAK7	CsWAKL17	0.002932	0.016705	0.175506
CsWAK19	CsWAKL7	0	0.002134	0
CsWAKL11	CsWAKL21	0	0	NaN

NaN; not a number, undefined/unpresentable.

**Table 3 plants-11-02703-t003:** qPCR primers list.

Gene Name	Forward qPCR Primer	Reverse qPCR Primer
CsWAK1	CAAGCCTCTCAAAGGAAGTATCTC	CATGAAAGAGCCCAATTAAGTCC
CsWAK4	TGCACTTGGTACAACAATTGG	GTTGTTGCAGTATTAACCCACC
CsWAK7	GGTTGCATAGATATTGATGAGTGC	GCACAAGGTTTCATTCTGTATTGG
CsWAKL1	GGATCTAAAGGAGATGGCACC	CCTATAGAGACACCCAAGGC
CsWAKL7	GAAGAACAAGGCAACTTACGGT	CCAAGAGCAGAACCTAAGCA
CsUBQ *	TACTGCGCCAGCTAACAAACC	GCACCCGTCTGACCTGAATC

* CsUbiquitin (CsUBQ) primers as the internal reference was retrieved from Guo et. al. (2018) [79].

## Data Availability

Not applicable.

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
