# Peer review of "Genome-Wide Identification and Expression Analysis of Wall-Associated Kinase (WAK) Gene Family in Cannabis sativa L."

_plants, 2022, doi:10.3390/plants11202703_

Round 1

Reviewer 1 Report

The article entitled, “Genome-Wide Identification and Expression Analysis of Wall- 2 Associated Kinase (WAK) Gene Family in Cannabis sativa L sum up the genome-wide analysis of the Wall- 2 Associated Kinase (WAK) Gene Family in Cannabis sativa. It is a regular analysis. Author needs to perform a few more analysis before acceptance of this Ms. Further, I could see several grammatical/typographical errors.

  1. How did the author confirm the number of WAK genes?
  2. Abstract should be revised. Sentences can not start with numbers. Line 14, how genes suddenly became 23 from 53?
  3. The authors should explain the significance of phylogenetic analysis.
  4. Protein protein interaction analysis and miRNA interaction analysis  should be the part of Ms. Author may refer the suggested Ms for the analysis. 
  5. Evolutionary analysis using Ka/Ks should also be included 
  6. Figures are not clearly visible.
  7. What is the significance of Motif analysis, if the importance of motifs has not been discussed.
  8. Plant materials or material?
  9. aeroponic conditions should be explained in details.
  10. Why RT PCR was conducted for only 5 genes? Atleast 10% genes could be analysed.
  11. How authors collected 100mg of trichomes? How many plants were used? How they are sure about the purity, nothing has been explained. 

Reviewer 2 Report

â‘ Two citations to the same reference on lines 35-38 can be abbreviated as one citation.

â‘¡Latin names of some species are not italicized. Such as line 45 in the text and line 590 in the references. Other places need to be checked again.

â‘¢You can add a table with information about the genes in the gene family.

â‘£The picture clarity needs to be improved, and some even seem to be cut directly from the website. All pictures should be checked.

⑤There are some problems with the evolutionary tree in FIG. 2 and FIG. 6. Some of the digital typesetting in FIG. 2 is out of place. It's even marked on the branches of the evolutionary tree. The kinship coefficients of the same proteins in the two graphs are not united. Such as CsWAKL15, CsWAKL28, CsWAK6, CsWAK11, CsWAK12.

â‘¥The FIG. 4 can be optimized and more beautiful.

⑦The legend in FIG. 6 is not clear, and the signified of (a), (b) and (c) are not indicated in the FIG. 6.

â‘§Collinearity in FIG. 5 can be increased for other species, with only one species being too few for Arabidopsis. And the name in FIG. 5 has not been changed to the new name in the article.

⑨The selected genes of the tissue expression can be used to research the function of the pathogen response you referred at the beginning of the article.

Round 2

Reviewer 1 Report

The interaction analysis can be done by selecting the orthologous sequences from other plants like arabidopsis, while for miRNA interaction, you need to upload both miRNA and gene sequences in psRNATARGET tool. If required you can see these Ms for further clarifications. https://www.sciencedirect.com/science/article/pii/S009884722100188X 

https://www.mdpi.com/2075-1729/12/7/941

The conclusion should not be a repeat of the abstract. It should be rewritten. 

Reviewer 2 Report

I think the revised MS could be accepted.

Author Response

Dear Reviewer,

I have attached the revised manuscript. All revisions to the manuscript have been marked up using the “Track Changes” function be easily viewed by the editors and reviewers. We have added protein-protein interactions and identifications of miRNA target.

  • Protein-protein interaction analysis was conducted using orthologous sequences from Arabidopsis. This has been addressed in line 290-314 2results), 428-448 (discussion) and 556-561 (material and methods).
  • miRNA sequences for Cannabis sativa were retrieved form Das et. al. (2015) and Hasan et. al (2016). Then, miRNA and CsWAK/CsWAKL transcript sequences were uploaded to PsRNATarget tool. And target sequences were detected and added to the manuscript as a supplementary table 3. The results, discussion and methods of miRNA analysis been addressed in line 315-325, 447-468 and 562-565. However, miRNA interactions could not be generated using Cytoscape software.  Because Cytoscape needs miRBase accession number to detect the interaction. But miRBase accession numbers are not available for C. sativa in Cytoscape. Precursor miRNA, hairpin and mature sequences should be emailed to mirbase@manchester.ac.uk after manuscript acceptance, and then the accession number can be taken. But this has not been done before. Das et. al. (2015) did not publish all the sequences, including hairpin, in their article, but shared some figures for pre-miRNA sequences not word or excel file and only some mature miRNA sequences are given as a supplementary table. In short, miRNA interactions could not be carried out because the miRBase accession number is not available for csa-miRNAs,

Our manuscript has undergone extensive English revisions by Prof. Dr. Gerald Berkowitz (author).

Regards

Round 3

Reviewer 1 Report

Accepted